# The Flipped Break-Even: Re-Balancing Demand- and Supply-Side Financing of Health Centers in Cambodia

**DOI:** 10.3390/ijerph20021228

**Published:** 2023-01-10

**Authors:** Sokunthea Koy, Franziska Fuerst, Bunnareth Tuot, Maurice Starke, Steffen Flessa

**Affiliations:** 1Improving Social Protection and Health, Deutsche Gesellschaft für Internationale Zusammenarbeit (GIZ), Phnom Penh 120102, Cambodia; 2Department of Health Care Management, University of Greifswald, 17487 Greifswald, Germany

**Keywords:** value-based healthcare, demand-side financing, supply-side financing, efficiency, Cambodia, costing

## Abstract

Supply-side healthcare financing still dominates healthcare financing in many countries where the government provides line-item budgets for health facilities irrespective of the quantity or quality of services rendered. There is a risk that this approach will reduce the efficiency of services and the value of money for patients. This paper analyzes the situation of public health centers in Cambodia to determine the relevance of supply- and demand-side financing as well as lump sum and performance-based financing. Based on a sample of the provinces of Kampong Thom and Kampot in the year 2019, we determined the income and expenditure of each facility and computed the unit cost with comprehensive step-down costing. Furthermore, the National Quality Enhancement Monitoring Tool (NQEMT) provided us with a quality score for each facility. Finally, we calculated the efficiency as the quotient of quality and cost per service unit as well as correlations between the variables. The results show that the largest share of income was received from supply-side financing, i.e., the government supports the health centers with line-item budgets irrespective of the number of patients and the quality of care. This paper demonstrates that the efficiency of public health centers increases if the relevance of performance-based financing increases. Thus, the authors recommend increasing performance-based financing in Cambodia to improve value-based healthcare. There are several alternatives available to re-balance demand- and supply-side financing, and all of them must be thoroughly analyzed before they are implemented.

## 1. Introduction

The majority of low- and middle-income countries have implemented a healthcare financing system where the government provides the resources required for healthcare facilities. Buildings are established and equipment is bought by the government, and the staff is hired directly by the Ministry of Health (MoH). Drugs are purchased by the MoH and provided free of charge to the facility [1]. During the 1960s and 1970s, almost 100% of the resources contained in government health facilities came directly from the government and patients did not have to pay formal fees for services delivered. In the 1980s, several countries introduced direct patient fees (out-of-pocket payments) [2], but these co-payments covered only a small percentage of the actual costs related to services delivered. In principle, healthcare financing in the public sector is still, to a large extent, supply-side financed by the government in many countries.

It was expected that patients would seek healthcare in these free-of-charge or highly subsidized facilities so that “universal health coverage” (UHC) [3] could be achieved. However, thorough analyses of country studies (including Bangladesh, Brazil, Ethiopia, Ghana, Indonesia, Thailand, and Vietnam) have demonstrated that a major portion of the population has no access to professional healthcare and many needs of the population remain unmet [4].

In some cases, patients cannot seek healthcare because the distance between their home and the closest healthcare facility is too great, especially for severely sick patients. In addition, household costs were higher than expected and prevented many patients from taking advantage of the free healthcare services offered. Transport costs pressured poor households and indirect costs (e.g., loss of work, harvest) prevented patients from seeking care. Sometimes it was also obvious that the quality of services of government healthcare services was too low and people preferred to go to private providers, even if they had to pay. Quality thus became a major factor for those seeking healthcare [5]. To summarize: Many patients worldwide face challenges in receiving value for their money concerning healthcare [6,7].

The relationship between cost (for society, for the financer, for the provider, for the patient) and value (e.g., quantity and quality of services, outcomes for the patient, impact on society) has become a major dimension of health economic research since Porter and Teisberg focused on “value-based healthcare” (VBHC) in their book “Redefining health care” [8,9]. They translate the efficiency (*E*) formula:(1)$$E=\frac{results}{resources}\to Max!\;$$
specifically for healthcare.
(2)$$E=\frac{health\;outcomes}{cost}\to Max!$$

This requires determining what the resources and respective results are. Figure 1 shows that resources are the consumption of the agents of production. They can be expressed as cost, accessibility, or components of structural quality. The healthcare institution or program transforms these agents into outputs, e.g., services and result quality. The individual (patient) consumes these outputs and produces their individual health as an outcome. Typical measures for it are morbidity, mortality, or quality of life. Finally, the health of the individual has an impact on the wealth, growth, security, and cohesion of the society. Consequently, any analysis of efficiency and/or value-based health care requires an assessment of the consumption of agents of production (usually expressed in their financial terms as cost) and the results they focus on.

Some authors reduce VBHC to a specific financing mechanism (e.g., performance-based payment [11,12]), but based on the efficiency formula, the starting point for all value-based healthcare is the knowledge of the cost of providing healthcare services as well as the knowledge of the health outcomes for the patient, i.e., the demand-side perspective is essential for value-based healthcare.

Many low- and middle-income countries have introduced some elements of demand-side financing in the new millennium [1]. These instruments included conditional cash transfers (for instance for preventive services) [13] and the reimbursement of transport costs for patients and their relatives [14]. At the same time, the government (or donors) introduced systems to refund fees paid by the patient. This did not only have an impact on the solvency of the private household but also changed the role of the patients as customers, particularly the poor. Patients became customers with a right to receive services.

Typical instruments of demand-side financing are [15]:Social health insurance: An (obligatory) social health insurance spreads the risk of illness among a major population. Members are entitled to appropriate treatment. The health facility receives a rebate for services [16,17];Community-based health insurance (CBHI): A CBHI is a small-scale voluntary instrument for the pooling of healthcare risks. It also refunds facilities for services provided [18,19];Vouchers: A voucher entitles the holder to a specific service (e.g., safe delivery). The voucher management agency refunds the provider for the services and is itself re-financed by the government and/or a donor [20,21];Funds for the vulnerable: In some countries, governments (or donors) provide special support for vulnerable groups, such as the poor, chronically ill patients, or the disabled. For instance, Cambodia Health Equity Funds (HEFs) entitle selected poor groups to receive free healthcare services. The provider is refunded by the HEF which itself is re-financed by the government and/or a donor [14,22].

The principle of these instruments is always the same: The customer is entitled to receive a service without out-of-pocket payment while the healthcare facility receives funds for treating the patient. The more patients they treat, the more funds they receive. In this way, facilities have the incentive to treat patients well so that they return and/or recommend the services of the institution to others. The management of the facility has (at least some) freedom to use the funds obtained from demand-side financing and becomes more active than it could be by waiting for supply-side financing. At the same time, demand-side financing historically requires strong institutions, is resource-intensive, and sometimes overburdens the limited capacities of low-income countries.

Most countries have a mixture of demand- and supply-side financing. German hospitals, for instance, receive all funds for buildings and equipment from the state government (supply-side) while ongoing expenditures are covered by health insurance schemes that pay only for services provided (demand side) [23]. The proper balance between demand- and supply-side financing must be determined for each individual country based on its capacity, population, and history.

In this paper, we focus on Cambodia, whose government started with 100% supply-side financing of hospitals and health centers. In 1997, direct patient fees were introduced which led to an exclusion of the poor [24]. Consequently, a HEF was established in 2000 to cover the poor [14]. Poor households were determined eligible for these funds through a national poverty identification system (“ID-poor”) [25]. HEF paid direct patient fees for its beneficiaries, which significantly increased access for this target group (Annear et al. 2019). However, the vast majority of the near-poor and other at-risk groups remained uncovered. For example, garment factory workers earned too much to be identified as poor but could not afford healthcare services if they fell sick. Consequently, a National Social Security Fund (NSSF) (officially launched in 2007) covering health insurance for the formal sector was launched in 2016. This helped steadily increase healthcare coverage of the population but mainly protected formal workers in the major cities of the country [26,27]. Thus, the country has started a transition towards demand-side financing, but it is unclear how far it has progressed on this pathway.

Most Cambodians still live in rural areas (61%; Census 2020) and work in subsistence farming or in the informal sector. In private health facilities, patients must pay direct fees out-of-pocket, which recover the full costs of the private providers. This indicates that the private facilities are demand-side financed. Public facilities, however, remain the main provider for those vulnerable groups, but the financial situation of public health centers is grossly unknown as the supply-side system of the government is located on different levels (MoH, Provincial Health Departments, Operational Districts, facilities) without a comprehensive picture of the flow of funds. The relevance of demand-side financing for the operation of these health centers is also largely unknown. It is necessary to assess the consequences of the current ratio of demand- and supply-side financing of health centers and what impact this ratio has on the quality of services as well as the efficiency of the providers. Furthermore, it is crucial to assess the impact of demand- and supply-side financing on equity (e.g., financial and spatial access to health centers). However, this analysis was already performed by other authors [14,25] while the impact of financing mechanisms on efficiency remains a research gap.

In this paper, we aim to fill this gap and analyze the consequences of the current balance between demand- and supply-side financing for primary healthcare centers in two provinces in Cambodia (Kampot and Kampong Thom). In the next section, we introduce the methodology of this analysis with a focus on a financial analysis of the facilities. This includes a “flipped break-even-analysis”. The respective results are demonstrated in the third section. These findings are discussed in the last section and include recommendations for the proper balance between demand- and supply-side financing.

## 2. Materials and Methods

### 2.1. Setting

Cambodia is a country in Southeast Asia with a total population of about 15.6 million [28]. During the last few decades, relevant health-related indicators have improved significantly [29]. For instance, the under-five mortality rate decreased from 124 deaths per 1000 live births (1995–1999) [30] to 16 (2015–2019) [29]. Infant and neonatal mortality rates have similarly declined. The maternal mortality rate (MMR) has fallen from 472 per 100,000 live births (2005) to 206 (2010), 170 (2014), and 154 (2021) [29].

At the same time, the utilization of public healthcare services has also increased noticeably. For example, the percentage of women who received antenatal care for their most recent live birth in the two years preceding the survey increased from 39% (2000) to 99% (2021), while the percentage of women who had four or more antenatal care visits increased from 9% (2000) to 86% (2021) [29]. The percentage of live births that are assisted by a skilled provider has gone up noticeably over the past two decades, from 34% in 2000 to 99% in 2021–2022 [29].

The Royal Government of Cambodia launched a series of healthcare system reforms in the 1990s including the introduction of cost-sharing (i.e., direct patient fees) at facilities and the provision of health services at the national, provincial, district, and health center level [31]. In 2021, there were 1250 public health centers, 94 district referral hospitals, 25 provincial referral hospitals, and nine national hospitals [30]. The health center (HC) is the frontline healthcare facility that provides the minimum package activity (MPA). This makes the HC the first contact point for people seeking healthcare. Normally, an HC covers around 8000 to 12,000 people. The HC mainly receives funds, human resources, buildings, equipment, drugs, and other materials directly from the government. Most health centers have a workforce of about 7–10 staff members with a focus on nurses or nurse-midwives. Very few health centers have a medical doctor or medical assistant. A special category exists, called a “health center with beds”, which provides MPA services and additional inpatient services. The number of staff at an HC with beds is usually higher than at health centers without a bed.

Appreciating that efficiency is a prerequisite to providing value-based healthcare for the population of Cambodia, the government decided to do a costing of healthcare services in the provinces of Kampot (coast) and Kampong Thom (center of the country). The costing was performed by GIZ (Deutsche Gesellschaft für Internationale Zusammenarbeit, Phnom Penh, Cambodia) which has a long-standing relationship with these two provinces. A sample of two operational districts was selected in each province and all health centers were analyzed in each OD. Table 1 shows a sample of the survey.

### 2.2. Financial Analysis

The financial analysis follows a standard procedure of healthcare administration applied in other studies in low- and middle-income countries [32,33,34,35]. All data were collected for the financial year of 2019 in the period 2020/21 by a team of trained data collectors under the leadership of the first author. Data were collected in Riel and United States Dollars (USD) but entered in USD ($1 = 4000 Riel).

The first element considered was income analysis. We distinguished between fees, government grants, donations, and other sources of income. The fees consisted of direct patient fees paid out-of-pocket by patients, payments made by the HEF (covering the poor groups identified in the ID-Poor Programme), rebates from the NSSF funds (covering the insured patients in the formal sector), voucher schemes (for special services, such as treatment of cervical cancer), other insurance schemes (such as community-based insurances), and other patient income. The government grants were partly given in cash, but the government also directly pays for staff (salaries and wages), drugs, and infrastructure investments (equipment, buildings, vehicles). In the latter case, the respective depreciation charge per annum of the investment was used as an annual government grant. Other government income was given as staff allowances (overtime, mid-wives incentive scheme). The respective figures were retrieved from accounting books in the health centers, the operational districts (OD), the provinces, and the MoH. In addition, figures for donations were sought from major donor agencies, such as UNICEF.

The second element was a traditional full cost calculation consisting of the analyses of cost categories, cost centers, and costing units. Following the governmental standards of public accounting, we distinguished between labor costs (salaries and wages for all personnel categories, social contributions, training, and other staff costs), costs of materials (drugs, medical materials, vaccines, other supplies), transportation costs, capital costs, and other expenditure (electricity, water supply, postage and telephone, printing and stationary, and sundry expenses). We also added costs for exemptions of direct patient fees and for transfers to the government. The capital costs of items > $1000 were depreciated by a straight-line method with national standards of length-of-life of equipment, buildings, and vehicles.

The cost of each cost category was allocated to the following cost centers: administration, outpatient department ((OPD), all curative services except for chronic patients), maternity (deliveries), services for chronic patients (patients requiring three or more contacts for their condition, e.g., for HIV, TB, diabetes, or hypertension), preventive services (vaccinations), and “other services” (antenatal care (ANC), post-natal care (PNC), family planning (FP)). The respective allocation of administration costs (as service cost center) to the other departments (final cost centers) was based on the number of cases using a simplified step-down approach. Consequently, we received the following costing units: cost per curative service (incl. minor surgery), cost per delivery, cost per contact with patients with chronic diseases (e.g., TB, diabetes, and hypertension), cost per preventive service (e.g., immunization), and cost per other services (ANC, PNC, family planning).

The step-down costing approach was used in similar studies in low- and middle-income countries [32,36,37]. It uses for each individual institution the actual figures, including service units per service center so that the respective service-mix is represented in the computation of unit costs. The respective manual for costing health centers in Cambodia can be obtained from the authors.

### 2.3. Break-Even-Analysis

The results from the financial analysis are used to calculate the surplus or deficit and the respective break-even point of the health center. This point represents the number of service units that allows the health center to avoid a loss. The respective break-even-point is calculated as shown in Equation (3):(3)$$e^{*}=\frac{C_{f}-I_{f}}{i_{v}-c_{v}}$$
with
*e**break-even-point [number of service units]*I_f_*fixed income*C_f_*fixed costs*c_v_*variable cost per contact*i_v_*variable income per contact

Figure 2 confronts the “normal” and the “flipped” break-even-analysis. Most textbooks [38] assume that fixed income is very small while fixed cost is high. At the same time, variable costs are less than variable income. Consequently, the profit zone starts to the right of the break-even point (BEP). In the flipped situation, fixed income is higher than fixed cost and the variable cost is higher than the variable income. Consequently, the loss zone starts to the right of the BEP.

We calculated the variables *I_f_, C_f_*, *c_v_*, and *i_v_* as averages for all the health centers considered in the study. Costs of drugs, medicine, and vaccines are assessed as variable. The same goes for income from direct patient fees, HEF, NSSF, and vouchers, while the other items are fixed costs or fixed income.

### 2.4. Quality Assessment

The quality of each health center was assessed by using the National Quality Enhancement Monitoring Tool (NQEMT). It is based on the Donabedian model focusing on structural, process, and result in quality [40], although it does not rigorously comply with the differentiations of Donabedian. The components of the structural quality are shown in Appendix A and include indicators such as the number and qualification of staff, the availability of buildings and equipment as well as the existence of certain programs (e.g., infection control) and administrative standards (e.g., for health equity fund management). For each component, respective indicators were defined. The process quality focused on the knowledge and skill of health workers. For this purpose, staff working at the OPD and maternity of the health center were randomly selected to assess their skills. An OPD case (clinical vignette) was presented, and the co-workers had to perform a role-play (e.g., “NeoNatalie”) so that their ability to handle relevant cases can be determined. The resulting quality relates to interviewing randomly selected clients (two patients from OPD and two from maternity) to assess client satisfaction. Each indicator was assessed against a certain standard (e.g., availability of a certain piece of equipment), and the scores were weighted, i.e., 30% of the total score was given to indicators of structural quality, 60% to process quality, and 10% to result in quality. In total, the maximum score was 100% and the minimum score was 0%. The quality assessments of health centers were conducted on a quarterly basis by external governmental staff, e.g., from the operational districts (OD) or from the Provincial Health Department.

Consequently, this study did not collect prime-quality data but used existing national quality statistics. These data were mainly based on structural parameters and outputs (as results), but did not include outcomes (e.g., equity, accessibility, services for key risk populations) or impact (e.g., health of population, economic potential), which would be highly relevant for the assessment of value-based health care. However, these data are unfortunately hardly available in Cambodia and/or cannot be referred to individual health facilities and their quality.

The relevant data for this analysis were presented in Appendix A. Unfortunately, the Ministry of Health of the Government of Cambodia assesses details of the quality assessment tool as confidential, i.e., merely the final score is accessible, but not the scoring for each indicator.

## 3. Results

### 3.1. Income Analysis

Figure 3 and Table 2 show that 93.93% of the total income for health centers is contributed by the government, while 6.07% is provided by fees (3.76% direct patient fees, 1.99% HEF, 0.27% NSSF, and 0.05% other). The other components (donations and other income) are negligible. On average, a health center has an annual income of $80,367.24, but there is a wide variation between $36,489.58 and $172,250.28. Three health centers that had been Level 1 hospitals before (CPA-1) were downgraded to health centers with beds. They have the highest income with $172,250.28, $111,321.89, and $109,249.02. Nevertheless, even without considering these three health centers, the differences are tremendous, especially when considering that theoretically, all health centers should have a similar structure, catchment population, and income.

Based on this analysis, we can state that merely 6.07% of the total income of health centers are demand-side financed. In fact, 61.94% of patient income comes from direct patient fees, 32.84% from HEFs, 4.38% from the NSSF, and 0.85% from other sources.

However, even these demand-side funds cannot be used by the public health center autonomously. One percent of the direct patient fees and NSSF revenues must be transferred to the national treasury, and 60% must be used for staff incentives. The leadership of the public health center only has decision-making power over the remaining 39% of the income from direct patient fees and NSSF revenues, which can be used for any operational expenses which the leadership of the facility assesses as most relevant. Even the revenues from the Health Equity Fund are not at free disposal. Sixty percent of HEF revenues are earmarked for staff incentives and 40% are available for free disposal by the facility leadership.

The largest portion of the government income is independent of the services provided, i.e., salaries and wages for staff (43.87% of government income), investments (4.40%), and other fixed lump sum payments (6.91%). A smaller portion is performance-based, such as (mid-wife) allowances (9.99% of government income) and the payment of essential drugs, medical materials, and vaccines, which are paid by the government according to consumption (34.82% of government income).

### 3.2. Cost Analysis

The average annual expenditure per health center is $79,889.89 with a range from $26,218.35 to $179,030.38. Again, the three health centers with beds have the highest cost. The largest portion of expenditures is salaries and wages at 54%, followed by drugs, materials, and vaccines at 35%. As Figure 4 shows, the total cost and the components are quite different between health centers. In particular, the cost of drugs and medical materials differs strongly. While the average health center spends some 35% of total expenditure on this category, one facility requires only 11% and another 55%.

The cost per service unit depends on the specific department and health center. On average, one outpatient department visit (general consultation) costs $7.90 with a range of $2.94 to $19.66. The cost per delivery is on average $149.11 with a maximum of $949.93. Contacts for a chronic patient cost on average $46.96 ($5.17–$257.06), contacts for prevention services cost $7.79 ($2.40–$38.38) and contacts for other services cost $8.47 ($0.89–$32.42).

Sixty-five percent of total costs are fixed, while the costs for drugs, materials, vaccinations, and mid-wife allowances are seen as variable, i.e., they are a function of the number of patients.

### 3.3. Surplus, Deficit, and Break-Even-Analysis

The respective figures for fixed income, variable income per service unit, fixed cost, and variable cost per service unit can be used to calculate the break-even point and compare it with the surplus or deficit of the institutions. On average, a health center makes a small surplus of $1533.01 with an actual number of service units of 12,316. The respective break-even point is calculated as 13,312.75. As Figure 5 shows, the few health centers with a loss are those that produce more service units than the BEP.

In order to demonstrate the impact of a changing workload we calculate the cost and income functions of an “average health center” assuming that financial statistics (fixed income, fixed cost, variable income, variable cost) are the respective averages as well as the number of service units are the averages of all health centers in the sample. These formulae must not be taken as standards for an individual facility as deviating service-mix will result in other functions. We can, however, show the general impact of altered workload on cost, income, and break-even point based on functions. The respective cost (*C*) and income (*I*) functions for health centers can be described by Equations (4) and (5):(4)C=50665.49+2.89·x
(5)I=75686.98+0.42·x
where *x* denotes the number of service units of the health center. For any health center in this sample, the fixed income is higher than the fixed cost while the variable income is lower than the variable cost per service unit.

### 3.4. Quality and Efficiency

The efficiency of a health center can be assessed by comparing the cost per service unit (input) with the quality score (output) of the respective center. Figure 6 shows the respective results. The average cost per contact is $6.67 with a variation coefficient (standard deviation/arithmetic mean) of 0.59. For the quality score, the respective figures were 0.74 and 0.12. The costs per service unit deviated more than the quality score but we still found a wide range of quality scores between 0.49 (Health Center 26) to 0.92 (Health Center K4). The average efficiency is 0.14, i.e., 0.14 quality points can be produced for $1 with a minimum of 0.03 (Health Center 33) and a maximum of 0.32 (Health Center 8).

Health Centers 8, 16, and K4 form the efficiency frontier under the assumption of variable or decreasing returns to scale. All other facilities either spend too many resources for a given quality score or produce too little quality for the resources consumed. Facilities on this efficiency border (or close to it) can work as benchmarks. The inefficient facilities could learn from the more efficient facilities about how to improve their production while maintaining quality. For instance, Health Center 8 uses $2.58 to produce 0.83 quality points, while Health Center 36 consumes almost exactly that amount ($2.59) but harvests a quality score of only 0.62. Facility 36 could benefit from a thorough analysis of what Facility 8 is doing differently.

## 4. Discussion

### 4.1. Health Facility Financing

The results of the study presented in this paper indicate that public health centers in Cambodia have different sources of funding. As Table 2 shows, 93.94% of total funding comes from the government as supply-side funding, and 51.83% of the total income is independent of the performance of the facility, as they arrive as a fixed line-item budget and represent equipment or staff paid for by the MoH. In addition, 32.71% of the total income is provided by the government in the form of essential drugs, medical materials, and vaccines. The facilities issue a monthly request to the operational district based on the average monthly consumption of the last three months, i.e., this element of supply-side financing is quantity-based. The operational district orders from the central medical store (CMS) in Phnom Penh for all of its health centers. The CMS does not have a monitoring and evaluation system to check whether an individual health center does indeed have the number of patients they declare and requires the respective drugs and materials. This task lies with the “Essential Drug Bureau” of the Drug Department of the MoH. There is a likelihood that drugs, materials, and vaccines are indeed performance-based while “performance” here means only the quantity of patients, not the quality of services.

The midwife allowances are also supply-side financed, but they are paid based on the number of live births at health facilities. Health facilities currently receive $15 per live birth delivery. Every month, individual health centers submit a report to the operational district. The reports are aggregated and submitted to the principal health department and then to the MoH. The reimbursed money is transferred directly to the account of the health center and the money is distributed among midwives. Consequently, the allowance is considered supply-side and performance-based, although “performance” refers merely to the number of deliveries, not the quality of treatment.

Generally (and not only for Cambodia as indicated in Table 2), healthcare financing is split into two dimensions. Firstly, we must discern the sources of funding. Income is received from the customer (patient) or from another source independent of the customer. The first type of income is called demand-side financing, which includes direct patient fees as well as third-party payers who pay the health facility for services rendered to the customer. The third party (in Cambodia: HEF, NSSF, and the voucher agency) pays the facility only if a patient receives healthcare services.

Secondly, income can be categorized as lump sum or performance-based. A lump sum payment is independent of any quantity or quality of healthcare services provided. For example, the government provides the buildings, equipment, and vehicles to healthcare facilities. It also employs staff based on the size of the population. Where applicable (e.g., faith-based health facilities beyond the scope of this paper [41]), donations are usually also lump sum payments. Whether patients seek healthcare services at the facility and whether they receive services there has no influence on the lump sum payment of the government.

However, patients, HEF, NSSF, and voucher agencies only pay if a patient receives treatment at the facility. This is performance-based funding. The respective criteria can be quantity (e.g., pay for number of outpatient visits) and/or quality-based (e.g., mark-up on the rebate if a certain quality indicator is reached such as low nosocomial infection rate or high patient satisfaction).

Internationally, many healthcare financing instruments have been developed (Figure 7). It is obvious that supply-side financing does not have to be provided as lump sum payments. The provision of drugs, materials, staff, equipment, buildings, etc., could also depend on the performance of the health facility, which could be measured in quantity (e.g., number of consultations) and/or quality. The latter is more difficult to assess and requires a comprehensive system of structural, procedural, and result-based indicators to measure quality [40,42]. The quality score, as described in Section 2.4, could be a starting point as an index number to justify and quantify performance-based supply-side facility funding, but many more parameters have to be considered, such as financial and spatial accessibility (equity).

In addition, demand-side financing offers a variety of instruments. Firstly, provider costs can be re-financed on the demand side. CBHIs, HEFs, social health insurance [14], and voucher agencies [43,44] pay the health facility only for pre-specified services. Again, this payment is mainly based on the number of services provided, such as deliveries, consultations, ANC/PNC contacts, etc. In principle, it could also be based on quality indicators.

Another aspect of demand-side financing is the refunding of direct household costs. For instance, households could receive transport vouchers or conditional cash transfers. This would increase the demand for healthcare services by reducing the direct cost of the private household. Finally, opportunity costs of the household, such as loss of labor and income can be covered by the government or donors. This increases the demand for healthcare services by reducing the opportunity costs of the household.

Several studies have shown that there is no silver bullet for healthcare financing. Neither demand-side nor performance-based financing guarantees efficient and equitable healthcare [27,45,46]. Even systems where patients do not have to pay direct fees and receive subsidies for direct household costs do not guarantee that all needs are met [14,25]. Other barriers, such as poor quality of services, low autonomy of health facility managers (resulting in the poor motivation of staff), and long distances might prevent people from seeking healthcare. Healthcare financing is an essential instrument in the struggle for universal health coverage, but not the only one.

Providing healthcare facilities the right to use their respective funds autonomously is crucial for effectively ensuring that healthcare financing creates successful healthcare services. The World Health Organization calls for higher autonomy in the way “in which health facilities and their broader provider and community networks receive, manage, and account for funds to deliver health services”. Health facilities should be established as “autonomous management entities, able to receive many types of funds directly and manage them independently to meet the needs of beneficiaries” [47]. Thus, healthcare financing is not only an issue of funding, but also of management, management training, and decentralization.

The cost and income functions shown above indicate that most health centers in the sample could increase their workload without running into a loss. Thus, the centers could handle more patients and higher demand. However, in the current setting, health centers have no financial incentive to improve their performance. Consequently, we can conclude that treating more patients brings them closer to a deficit, and better quality is not financially rewarded. If patients do not provide money via fees (demand-side financing) and the government does not reduce lump sum line-item funding and increase performance-based funding, we cannot expect that patients will receive value-for-money health care at Cambodian health centers. At least from a financial perspective, we must state that the patient is a disturbing factor in the system—more patients and more severe diseases correlate to a worse financial situation for the institution. This system does not, therefore, encourage improved performance.

### 4.2. Quality and Efficiency

Figure 6 shows that the efficiency (calculated as the quotient of quality score and cost per service unit) of health facilities strongly differs. This calls for an analysis of the reason behind this phenomenon. Based on the data presented in Section 3 and in Appendix A Appendix A, Table 3 shows the relationship between unit costs, quality, efficiency, and the so-called margin (*m*, variable cost minus variable income) with the variables declared above as shown in Equation (6):(6)$$m=c_{v}-i_{v}$$

As seen above, the variable cost is higher than the variable income, which leads to a risk of neglecting patients resulting in poor quality. There is a likelihood that a higher difference between variable cost and variable income (m) is associated with lower efficiency.

Table 3 shows that the unit costs are not correlated with the quality of healthcare services. One would expect that higher unit costs would lead to higher quality because investments in (structural) quality should lead to higher quality care. However, our data do not support this assumption. If the cost per service unit is not at all related to the quality of services, we can state that patients receive little value for their money.

The unit costs explain 66% of the total variance in the efficiency of the health centers. If the unit costs increase by $1 the efficiency decreases significantly by one percentage point. At the same time, efficiency has no significant influence on quality.

If we analyze the relationship between the margin and the quality, we received an unexpected result: a higher margin led to significantly better quality. If the difference between variable cost and variable income increased by $1, the quality score increased by two percentage points. However, the margin explained only 7% of the total variance and should, therefore, not be stressed too much. Even so, this result calls for proper quality management. Healthcare financing reforms that include a transition from lump sum fixed income to variable income need to be accompanied by instruments that ensure the quality of services is not negatively affected.

Instead, the margin explains 43% of the variance of the efficiency. A higher margin significantly reduces efficiency. Figure 8 shows the respective data and a best-fit power function (R^2^ = 0.54). It is obvious that an increasing difference between variable cost and variable income leads to lower efficiency of the facility. In other words, institutions that receive a higher variable income from patients or from the government have a clear incentive to work more efficiently. Consequently, facilities with higher income from direct patient fees, HEFs, vouchers, and the NSSF tend to be more efficient than facilities that are entirely budgeted by the government. At the same time, other studies have proven that direct patient fees induce a risk of inequity and prevent patients from seeking healthcare at all [14,48]. Our findings, in combination with these studies, call for a higher share of HEF, vouchers, and NSSF funding to increase the efficiency of rural health centers.

### 4.3. Limitations

The results presented here are subject to several limitations.

Quality of data: The collection of data was limited by the Cambodian reality of health facilities and public accounting. We had to use different sources of data, such as public accounts from operational districts, provinces, the MoH, and other organizations (e.g., central medical stores). Payments were accounted for in the place where they are made, and no reconciled accounts exist in Cambodia. In addition, some data had to be collected at the health facility directly from staff, which required personal visits. There was a risk that some data were not precise or might represent only the specific situation of the day of the visit;Timeliness: The data were from the financial year 2019. It took about 12 months until data were available after the end of the financial year, and the COVID-19 pandemic delayed some research visits. Thus, the data might not be representative of the current situation;Quality: We had to rely on the quality score provided by the National Quality Enhancement Monitoring Tool (NQEMT). This tool is mainly based on structural quality components and might not reflect the result quality sufficiently. This system does not sufficiently take into account the proportion of different services provided, i.e., the quality score cannot reflect the proportion of different services provided. It is possible that a health center could achieve higher quality by providing a higher proportion of less complex services. In principle, the regulations of the government describe precisely the services and service-mix of health centers, but in reality, we found diverging service statistics with an impact on quality score.Furthermore, the quality scores do not deviate very much. It seems that there is a tendency for quality assessment to give rather similar scores. Only one institution has a score higher than 90%, while only two have a score lower than 50%. Anecdotal evidence suggests that this assessment is partially biased;Outcome: Quality of services (and in particular structural quality) is only a prerequisite of good outcomes relevant for customers. Others are financial and spatial access, availability of key services, and coverage of key at-risk populations. With the limitations of data availability in Cambodia, we cannot add these additional dimensions.Sample: We must address that the sample was drawn from the provinces of Kampot and Kampong Thom. These provinces are not representative of urban areas (in particular Phnom Penh) and very remote areas with very low population density (such as Mondulkiri Province). Thus, any conclusions built on this sample are limited to these provinces.However, in comparison to many other costing studies done in low-income countries, the methodology and the data collection process are quite precise, and much effort was invested to safeguard good quality data so that the results presented above are comparably reliable;Average break-even analysis: Another methodological shortcoming is the fact that we developed a break-even analysis for an “average” health center. While we calculated the fixed and variable income and expenditure for each facility, the final result is an “average institution” with a fixed service-mix. This is appropriate to demonstrate the impact of increased workload, but it should not be taken as an accusation against a single health facility. In addition, our explanation that differences in surplus and deficit are mainly due to utilization disregards the fact that these differences could also be explained by differences in the severity of the case-mix;Efficiency: The efficiency statistic is built on a rather simple index number as the ratio between the quality score and cost per service unit. The number of service units was calculated as the total of service units for all cost centers. This is a simplification as some facilities might show poor efficiency while they are very efficient in one particular service. Other authors have shown that data envelopment analysis (DEA) can overcome this problem [49], an alternative could be stochastic frontier analysis (SFA). However, the focus of this paper is on the flipped break-even and the impact on the value-for-customer. It should, however, be respected that this limitation must not lead to an accusation against an individual institution. It should, however, open the platform for further discussion on quality, cost, and efficiency between the government and the leadership of health facilities;Optimum: The findings of this study demonstrate that Cambodia has a mixed system of lump sum line-item funding and performance-based funding as well as supply-side and demand-side financing of public health centers. The biggest share of income of a public health center is independent of any quantitative or qualitative performance, and the leadership of the facility has very low autonomy over its finances. Our findings cannot indicate an “optimum” ratio between the healthcare financing alternative, but by summarizing the findings, we can state that Cambodia should move towards performance-based financing to improve the value for the customer. There is still a need for line-item budgeting for preventive services and for health facilities in remote areas with a low population density. It might also be wise to provide buildings, equipment, and vehicles based on the population of the catchment area. The government should, however, consider providing funds for staff based on performance criteria.

The development of HEFs, the NSSF, and voucher systems is a step in the right direction to empower patients and overcome the hazard of a “flipped break-even”. Our results suggest that value-for-money for the patient and efficiency are more likely to be achieved if variable income is higher than variable cost. However, efficiency is only one dimension of value-based healthcare. With the scope of this study and with this methodology we cannot address, for instance, the personal value-for-money of the patient, which covers dimensions such as friendliness, acceptance, cohesion, etc.

## 5. Conclusions

The Royal Government of Cambodia has taken important steps towards universal health coverage by introducing the Health Equity Fund (HEF) and the National Social Health Insurance (NSSF). Based on the results of the costing of health centers in two provinces of Cambodia we conclude:The results of this paper indicate the relevance of regular routine data collection in the healthcare system of Cambodia, in particular costing and quality of healthcare services as well as their impact on equity and efficiency. This will build the evidence base for re-balancing demand- and supply-side financing and encourage value-based healthcare. Consequently, the results presented here encourage switching from a snapshot-like costing to a routine system based on national samples. In most countries, costing of healthcare services is accepted as a prerequisite of proper healthcare planning. The Royal Government of Cambodia recently presented the “Cambodia PHC Booster Implementation Framework” calling for professional costing in order to “adjust reimbursements rates to the comprehensive cost estimates which should be updated regularly” [50];The financial analysis demonstrates that merely 2% of the total income of public health centers comes from the HEF. This surprising statistic indicates that vulnerable groups who are entitled to healthcare without out-of-pocket payments do not seek healthcare at all or find ways to pay for private healthcare services. This goes in line with the findings of other studies showing that the HEF does not necessarily protect the poor as expected [51]. Some authors argue that the HEF card carries a stigma as it indicates that the cardholder is vulnerable [52,53], but our research cannot assess the relevance of this assertion. The under-utilization of healthcare services by the poor and the role of the private sector will require more research;The different instruments of healthcare financing presented in Figure 7 show that there are different financing options that can be mixed. For a political discussion, it is crucial to distinguish between the terms and precisely define the meaning of certain concepts. For instance, the terms “demand-side-financing”, “performance-based financing” and “household subsidy” are not strictly defined in Cambodia, leading to misunderstandings. Thus, our findings call for a standardization of terminology at least within the Royal Government of Cambodia and the health partners (donor agencies);The efficiency of the health centers can be improved by increasing the variable income. Currently, the public healthcare system is under-financed so that additional funds (e.g., from the National Social Security Fund) will not lead to overfunding of healthcare services. However, if the relevance of demand-side and/or performance-based financing increases we will have to formulate a transition strategy with a stepwise shift from fixed income towards variable income without double financing;The transition from supply- to demand-side financing has to be accompanied by quality management measures. Cambodia has embarked on a process for accreditation of healthcare facilities. This development should be continued and strengthened so that supply-side funding becomes more and more performance-based;The sample from Kampot and Kampong Thom provinces is not representative of very remote areas with low population density. In some areas, it is impossible to switch more towards demand-side financing as the demand of the small population is too low to safeguard that the facilities could survive without line-item lump sum funding. Public health centers are needed in certain places to allow for acceptable access time, but they will not survive from rebates of the HEF and the NSSF or variable income from the government. These facilities will require a fixed income for the foreseeable future. This calls for a more detailed analysis of specific situations. Again, routine accounting, costing, and quality assessments are prerequisites for these decisions;The description of the pathways of information, materials, and funds from the facility to the operational district and from the provincial health department to the MoH indicates that the system is highly complex, slow, and expensive. Systems like that tend to be prone to corruption and mismanagement. Cambodia has started a process of decentralization [54]. There is a clear need to delegate more responsibilities to lower levels, such as operational districts. Public health centers should receive a higher degree of autonomy in order to motivate staff and make better decisions reflecting the particular situation of this facility. Strengthening demand-side and performance-based financing is an important component of this process.

Thus, we can conclude that increasing the share of performance-based financing in Cambodia is recommended to improve value-based healthcare. This can be achieved in different ways. For instance, the government can shift its input to public health centers from lump sum to performance-based financing. At the same time, insurance and support systems (e.g., NSSF, HEF, and voucher agencies) should increasingly cover a higher share of the population, which could also be achieved by subsidizing these institutions with government budgets, which previously were given directly to public health centers as line-item budgets. There is a need to re-balance demand- and supply-side financing, but all of them must be thoroughly analyzed and implemented based on sound evidence.

## Figures and Tables

**Figure 1 ijerph-20-01228-f001:**
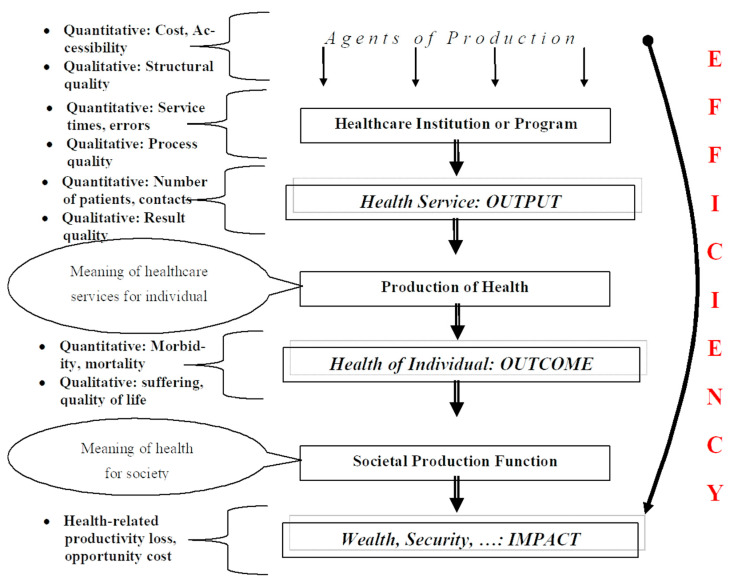
Health economic production and efficiency. Source: own, based on [10].

**Figure 2 ijerph-20-01228-f002:**
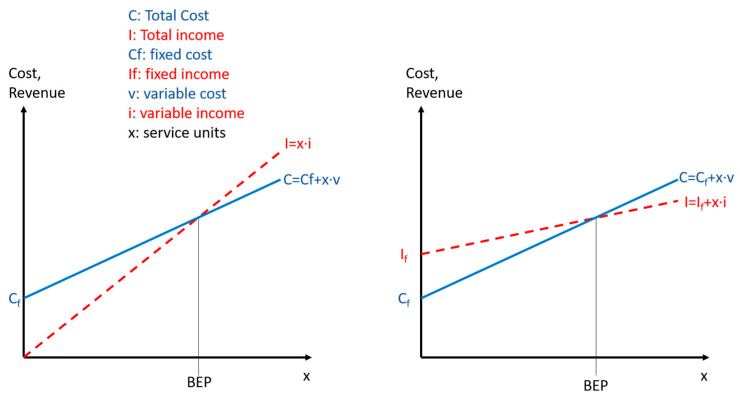
Traditional and flipped break-even. Source: own, based on [39].

**Figure 3 ijerph-20-01228-f003:**
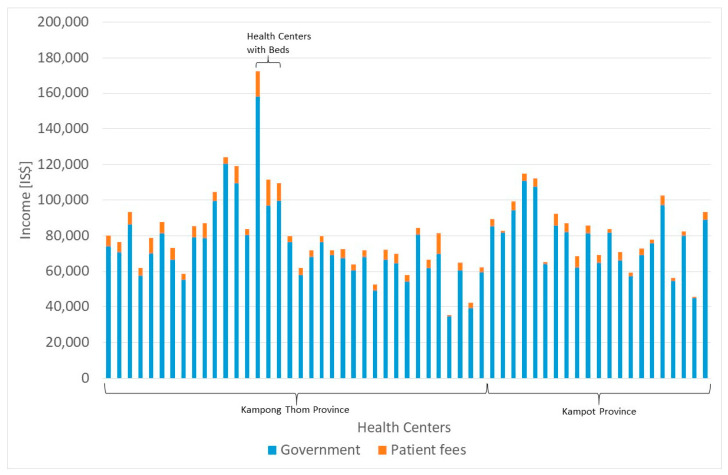
Income categories. Source: own.

**Figure 4 ijerph-20-01228-f004:**
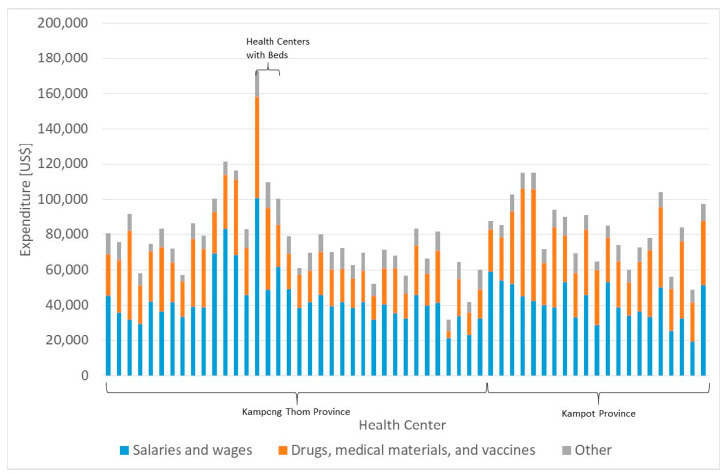
Expenditure categories. Source: own.

**Figure 5 ijerph-20-01228-f005:**
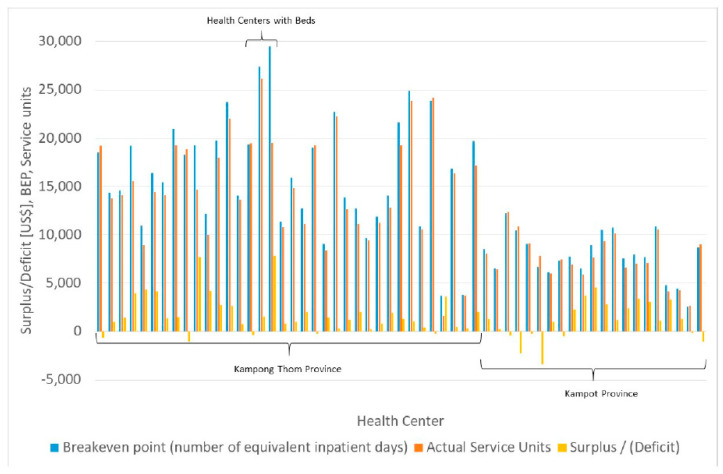
Break-even point, actual service units, and surplus/deficit. Source: own.

**Figure 6 ijerph-20-01228-f006:**
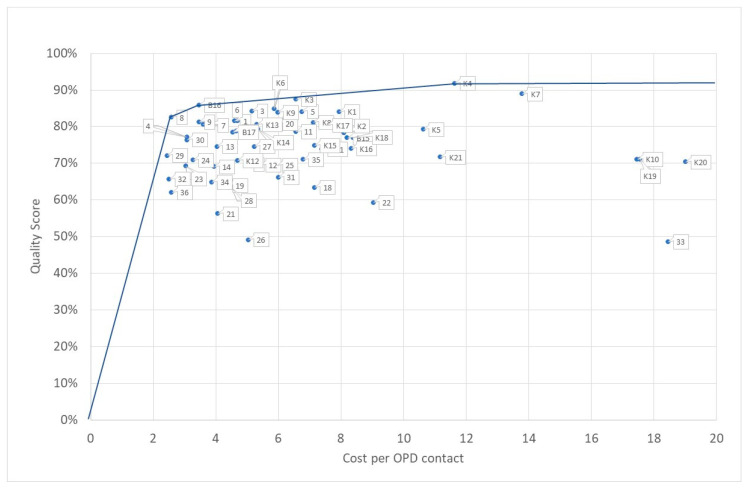
Efficiency of health centers. Source: own.

**Figure 7 ijerph-20-01228-f007:**
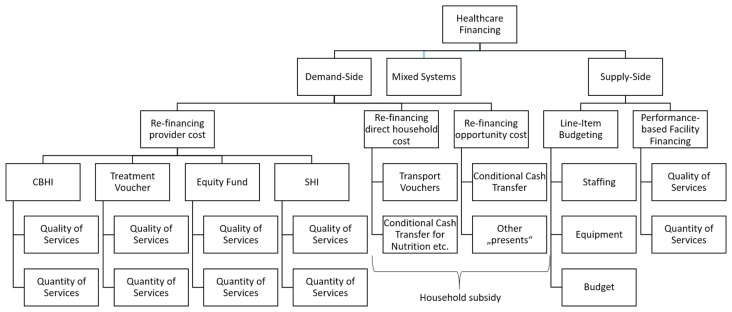
Instruments of healthcare financing. Source: own.

**Figure 8 ijerph-20-01228-f008:**
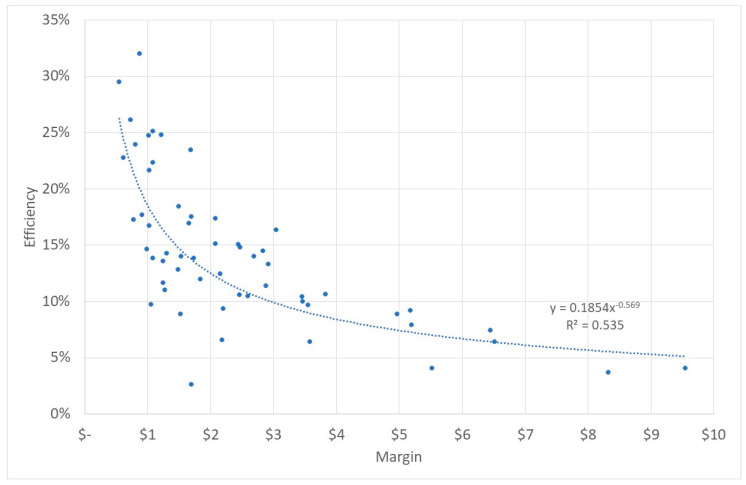
Margin and efficiency. Source: own.

**Table 1 ijerph-20-01228-t001:** Sample.

Provinces	Operational District	N	Name of Health Center
Kampot	Kampong Trach	12	Ang Sorphy, Boeng Sala, Banteay Meas, Prek Kreus, Sdach Kong, Srae Chea, Svay Tong, Tnoat Chong Srang, Prey Tonle, Phnom Lo Ngeang, Russei Srok Keut, Kampong Trach
	Angkor Chey	9	Ang Phnom Toch, Deum Dong, Samrong Leu, Champei, Wat Ang, Dan Koum, Sam Larnh, Pra Phnum, Trapaing Sala
Kampong Thom	Kampong Thom	17	Tboung Krapeu, Srayov, Prey Kuy, Damrei Choan Khla, Kampong Thom, Achar Leak, Ka Koh, Kampong Ko, Kampong Svay, Damrei Slab, Sandan (with beds), Chheu Teal, Mean Chey, Chhouk, Taing Krasao, Sala Visai (with beds), Sambor (with beds)
	Baray-Santuk	19	Ti Pou, Pra Sat, Kampong Thmor, L’ak, Thnoat Chum, Balangk, Chaeung Daeung, Krava, Beoung, Chhouk Ksach, Baray, Sralau, Kreul, Srah Banteay, Pratoang, Kork Trabaek, Pong Ro, Korki Thom, Chong Dong

**Table 2 ijerph-20-01228-t002:** Sources of funding of public health centers in Kampong Thom and Kampot Provinces (percentage of total income). Source: own.

	Demand-Side Financing	Supply-Side Financing	Total
Lump Sum Financing		Salaries and wages: 41.21% Investments: 4.13% Other lump sums: 6.49%	51.83%
Performance-Based Financing	Direct patient fees: 3.76% HEF: 1.99% NSSF: 0.27% Other: 0.05%	Drugs, materials, vaccines: 32.71% Mid-wife allowances: 9.39%	48.17%
Total	6.07%	93.93%	100%

**Table 3 ijerph-20-01228-t003:** Regression analysis (OLS).

Independent Variable	Dependent Variable	y-Intercept [*p*%]	Slope [*p* %]	R^2^
unit cost	quality	0.75 [<0.01]	0 [0.64]	0.00
unit cost	efficiency	0.23 [<0.01]	−0.01 [<0.01]	0.66
efficiency	quality	0.71 [<0.01]	0.22 [0.25]	0.02
margin	quality	0.68 [<0.01]	0.02 [0.04]	0.07
margin	efficiency	0.20 [<0.01]	−0.02 [< 0.01]	0.43

## Data Availability

Data available from the authors.

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
