# Peer review of "The Flipped Break-Even: Re-Balancing Demand- and Supply-Side Financing of Health Centers in Cambodia"

_ijerph, 2023, doi:10.3390/ijerph20021228_

Round 1
Reviewer 1 Report
General comments
This is a valuable contribution to the analysis of the financing of health services in LMIC, which provides some interesting findings in terms of the balance between supply side and demand side financing of health centres.
However, a major limitation of the study is the use of the quality assessment as the measure of outcome. While the paper does not clarify in detail the contents of the quality assessment, it does not appear to include key elements of the expected outcome of health services, such as the coverage of key health measures in the population. This is particularly relevant in the Cambodian context, as improving equity by improving coverage of key health measures, is an important objective for the health system.
A second potential limitation of the analysis is in the calculation of service costs which appears to assume similar service mix across health centres, while costs would differ if facilities offered differing mixes of services in different health centres.
If the authors could acknowledge and / or respond to these concerns, the paper would be acceptable for publication.
Specific comments by section
Abstract provides little information on data source or analysis method; no mention of dates of data used in analysis.
Introduction
Line 39. What type of countries do these analyses refer to ? ie is this limited to LMIC ? While LMIC are mentioned in the first line, it is not clear whether the subsequent statements also apply to LMIC.
Line 60 Its not clear how the efficiency formula quoted in line 57 relates to the comment that some element of demand side financing is required.
Line 100 ff provide a good outline of the development of the Cambodian health financing system and its current status, as well as the rationale for the study and its objective.
Line 126 consequences – the analysis focuses on the consequences for efficiency, but it is worth noting that most of the demand side financing strategies were introduced with the aim of improving equity, rather than efficiency. Balancing equity and efficiency remain key challenges for health system financing.
Materials and methods
Line 162 ‘sample of the survey’ – refers to a survey but this has not been described/ introduced. Eg ‘a survey was conducted in the following districts of two provinces’. It would be useful to explain here the rationale / reason for selection of these provinces / districts and the extent to which they might represent the health system.
Table 1 The names of the facilities probably do not mean much to the average reader. But what type of facility are these ? HC with beds / HC without beds; or hospitals ?
Financial analysis – no comments
2.4 Quality assessment. It is worth noting that the quality assessment is used as the measure of the outcome for the analysis. However, the indicators / measures of quality are largely based around inputs – buildings, equipment, health workers. The output element is only 10% and that is based on client satisfaction. In considering the strengths and weaknesses of the study, the quality assessment does not address elements such as the population coverage of key services, which represent important outcomes of the health service.
3. Results
Line 265 This seems to repeat the information in line 262 with the difference that 39% is mentioned in line 263, and 40% in line 266.
Line 278 refers to the cost of ‘drugs and medical materials’, while Figure 3 divides expenditure into salaries and wages, and ‘store’. It is not clear whether these two categories are the same, or whether ‘store’ refers to something in addition to drugs and medical materials.
Line 293-4 seems to repeat the phrase ‘fixed income, variable income per service unit’ when perhaps what is meant is fixed expenditure and variable expenditure per service unit in the second iteration..
Line 304. It is not clear how the differences in the proportion of services in different departments are addressed in calculation of the income per service unit. As noted in lines 283 ff, the cost per service unit varies with the department or type of service. However, the cost and income functions use only a single figure (x) for the ‘number of service units’. While comparing the fixed and variable income per service within a given health centre does not need to take this into account, comparison between different health centres needs to consider the mix of service types provided. For example, facilities that provide a higher proportion of high cost services (eg deliveries) will have a higher average cost per service unit
3.4. Quality and efficiency.
Line 310 The proposed definition of efficiency does not appear to take into account the proportion of different services provided, as noted above. While the cost per service varies with the proportion of different services, the quality score is not reported as reflecting the proportion of different services provided. The implications of this definition need to be considered. For example, it is possible that a health centre could achieve higher efficiency by providing a higher proportion of less complex, or less expensive outputs (eg fewer deliveries, more outpatient visits), while still achieving a high quality score.
It is unclear if the quality score takes into account differences in proportion of services provided. If not included, the authors might consider including some sort of weighting for the proportion of different services provided in calculating the number of service units. For example, using the number of ‘equivalent OPD visits’ for each service, based on cost relative to a standard OPD visit.
Alternatively separate costs per service unit and frontiers could be calculated for each of the key services provided, which may demonstrate that different facilities perform more efficiently on different services.
4. Discussion
4.1 Health facility financing.
Line 380 Use of the quality score as a measure of performance. As noted above, the quality score does not appear to include measures of the population coverage of services, and, in particular, the equity of that coverage. This is a significant limitation in the use of the quality score as a measure of outcome.
Line 416 – no incentive – but what is meant here is no financial incentive; there are other types of incentive.
4.2 Quality and efficiency
This section and the regression analysis appear to be additional analysis of results although the source of data for this analysis is not clearly stated. If this is further analysis of the data from the results section, it is unclear why this section is included under discussion, rather than results.
Line 436 See comments above on the calculation of unit costs and the need to consider the mix of service types in this calculation. Unit costs will also depend on the service mix, and thus service mix becomes an intermediate variable which could explain the lack of correlation between unit costs and quality score.
Line 444 and 452. In discussing the relationship between margins and quality or efficiency, it may also be relevant to consider the service mix, as margins may vary between services, with, for example, higher margins for services that contribute more to the quality score.
4.3 Limitations.
Given the reliance on quality scores to measure efficiency, the lack of deviation in these scores is of concern. A significant issue is the lack of information on how the quality scores are calculated, and to what extent quality scores may reflect or be influenced by the mix of services provided.
As noted above, use of the quality score as a measure of the outcome of health service provision does not appear to include key elements of the expected outcome, notably the coverage of key services, and particularly coverage of key at risk populations.
5 Conclusions
Line 571 ‘recommendable’ seems a rather oblique way of saying ‘is recommended’.
Reviewer 2 Report
The topic of the article is very interesting from both a research and a policy perspective.
My main comments are mainly about the methodology.
1. Paragraph 2.4. Quality. The article lacks a classic descriptive statistics table to show the dispersion of the values.
2. Paragraph 3.3. Same remark. Comments on lines 298-301. I think one should be more cautious, and the question arises whether the differences in surplus and deficit might not be explained largely by differences in the severity of the case-mix.
3. Equations A and 5. Has it been tested different functional forms, which is an important point.
4. Paragraphs 3.4 and 4.2. I don't understand how efficiency was calculated. Efficiency is the relationship between inputs and outputs. What are they? It is not clear to me. What is the method used? The most appropriate method would be a non-parametric method of the DEA type (without or with bootstraps) or Order-M for example, having previously checked for the ouliers, a method that allows several inputs and several outputs to be taken into account. It seems that this type of analysis has been used (lines 318 and 319), but it is not clear. This issue needs to be clarified and basic statistics on inputs and outputs as well as on scores need to be presented. The validity of the conclusions made in lines 413-440 seems extremely fragile until the methods used are clearly specified. This is unfortunate because there are still too few studies that highlight the potential room for efficiency gains when all health systems, including at the first level, are facing very severe resource constraints. These issues need to be addressed all the more rigorously because, as the authors point out, the samples are not randomized.
5. A table of descriptive statistics should be included for all variables analyzed in the paper.
6. It would be interesting to see if relatively homogeneous groups could appear (with an ascending hierarchical classification analysis for example).
7. The introduction, interesting in itself, is a bit long with several elements that are outside the central theme of the paper.
I encourage the author(s) to resubmit their paper taking into account the above remarks, because as I mentioned at the beginning of my comments, the theme treated is very interesting. The same is true of many analytical considerations.
Round 2
Reviewer 1 Report
The authors have responded in detail and with care to the comments on the first version of the paper. While this has increased the length of the paper, the additional information provides greater understanding of the context, and the data limitations which are necessary for the reader. With these revisions, the paper is acceptable for publication.
Author responses are described below in italics against the comments on the original versionl
Abstract provides little information on data source or analysis method; no mention of dates of data used in analysis.
Addressed – additional information provided
Introduction
Line 39. What type of countries do these analyses refer to ? ie is this limited to LMIC ? While LMIC are mentioned in the first line, it is not clear whether the subsequent statements also apply to LMIC
Addressed
Line 60 Its not clear how the efficiency formula quoted in line 57 relates to the comment that some element of demand side financing is required.
Addressed
Line 100 ff provide a good outline of the development of the Cambodian health financing system and its current status, as well as the rationale for the study and its objective.
Line 126 consequences – the analysis focuses on the consequences for efficiency, but it is worth noting that most of the demand side financing strategies were introduced with the aim of improving equity, rather than efficiency. Balancing equity and efficiency remain key challenges for health system financing.
Addressesd – additional comment on equity provided
Materials and methods
Line 162 ‘sample of the survey’ – refers to a survey but this has not been described/ introduced. Eg ‘a survey was conducted in the following districts of two provinces’. It would be useful to explain here the rationale / reason for selection of these provinces / districts and the extent to which they might represent the health system.
Addressed – additional information provided
Table 1 The names of the facilities probably do not mean much to the average reader. But what type of facility are these ? HC with beds / HC without beds; or hospitals ?
Addressed
Financial analysis – no comments
2.4 Quality assessment. It is worth noting that the quality assessment is used as the measure of the outcome for the analysis. However, the indicators / measures of quality are largely based around inputs – buildings, equipment, health workers. The output element is only 10% and that is based on client satisfaction. In considering the strengths and weaknesses of the study, the quality assessment does not address elements such as the population coverage of key services, which represent important outcomes of the health service.
Addressed – lack of outcome data acknowledged but agree that reliable data difficult to obtain.
3. Results
Line 265 This seems to repeat the information in line 262 with the difference that 39% is mentioned in line 263, and 40% in line 266.
Clarified
Line 278 refers to the cost of ‘drugs and medical materials’, while Figure 3 divides expenditure into salaries and wages, and ‘store’. It is not clear whether these two categories are the same, or whether ‘store’ refers to something in addition to drugs and medical materials.
Corrected
Line 293-4 seems to repeat the phrase ‘fixed income, variable income per service unit’ when perhaps what is meant is fixed expenditure and variable expenditure per service unit in the second iteration..
Corrected
Line 304. It is not clear how the differences in the proportion of services in different departments are addressed in calculation of the income per service unit. As noted in lines 283 ff, the cost per service unit varies with the department or type of service. However, the cost and income functions use only a single figure (x) for the ‘number of service units’. While comparing the fixed and variable income per service within a given health centre does not need to take this into account, comparison between different health centres needs to consider the mix of service types provided. For example, facilities that provide a higher proportion of high cost services (eg deliveries) will have a higher average cost per service unit
Further explanation of the use of the ‘average centre’ provided – this further clarifies the analysis; and has also been included in the limitations
3.4. Quality and efficiency.
Line 310 The proposed definition of efficiency does not appear to take into account the proportion of different services provided, as noted above. While the cost per service varies with the proportion of different services, the quality score is not reported as reflecting the proportion of different services provided. The implications of this definition need to be considered. For example, it is possible that a health centre could achieve higher efficiency by providing a higher proportion of less complex, or less expensive outputs (eg fewer deliveries, more outpatient visits), while still achieving a high quality score.
This issue is addressed by further comment in the limitations
It is unclear if the quality score takes into account differences in proportion of services provided. If not included, the authors might consider including some sort of weighting for the proportion of different services provided in calculating the number of service units. For example, using the number of ‘equivalent OPD visits’ for each service, based on cost relative to a standard OPD visit.
Alternatively separate costs per service unit and frontiers could be calculated for each of the key services provided, which may demonstrate that different facilities perform more efficiently on different services.
These issues are further noted in the limitations
4. Discussion
4.1 Health facility financing.
Line 380 Use of the quality score as a measure of performance. As noted above, the quality score does not appear to include measures of the population coverage of services, and, in particular, the equity of that coverage. This is a significant limitation in the use of the quality score as a measure of outcome.
Addressed – additional comment in limitations
Line 416 – no incentive – but what is meant here is no financial incentive; there are other types of incentive.
4.2 Quality and efficiency
This section and the regression analysis appear to be additional analysis of results although the source of data for this analysis is not clearly stated. If this is further analysis of the data from the results section, it is unclear why this section is included under discussion, rather than results.
Line 436 See comments above on the calculation of unit costs and the need to consider the mix of service types in this calculation. Unit costs will also depend on the service mix, and thus service mix becomes an intermediate variable which could explain the lack of correlation between unit costs and quality score.
Addressed by additional comment in limitations
Line 444 and 452. In discussing the relationship between margins and quality or efficiency, it may also be relevant to consider the service mix, as margins may vary between services, with, for example, higher margins for services that contribute more to the quality score.
4.3 Limitations.
Given the reliance on quality scores to measure efficiency, the lack of deviation in these scores is of concern. A significant issue is the lack of information on how the quality scores are calculated, and to what extent quality scores may reflect or be influenced by the mix of services provided.
As noted above, use of the quality score as a measure of the outcome of health service provision does not appear to include key elements of the expected outcome, notably the coverage of key services, and particularly coverage of key at risk populations.
5 Conclusions
Line 571 ‘recommendable’ seems a rather oblique way of saying ‘is recommended’.
Corrected
General comments
This is a valuable contribution to the analysis of the financing of health services in LMIC, which provides some interesting findings in terms of the balance between supply side and demand side financing of health centres.
However, a major limitation of the study is the use of the quality assessment as the measure of outcome. While the paper does not clarify in detail the contents of the quality assessment, it does not appear to include key elements of the expected outcome of health services, such as the coverage of key health measures in the population. This is particularly relevant in the Cambodian context, as improving equity by improving coverage of key health measures, is an important objective for the health system.
Response: Additional information provided in Methods section – addressed. The limitations arising from use of this data has been noted.
A second potential limitation of the analysis is in the calculation of service costs which appears to assume similar service mix across health centres, while costs would differ if facilities offered differing mixes of services in different health centres.
Response: additional information in regards to how service mix differences are accounted for in the costing has been provided – this has been addressed.
If the authors could acknowledge and / or respond to these concerns, the paper would be acceptable for publication.
Further information on the costing methods and use of the ‘average institution or facility’ as measure for calculation purposes.
With these and the other revisions, the paper is acceptable for publication.
Author Response
Thank you very much! We add here on Reviewer 2 because the system does not allow to do that there.
